# Joint Motion Estimation with Geometric Deformation Correction for Fetal Echo Planar Images Via Deep Learning

**Jian Wang**[*1]                                              JIAN.WANG@CHILDRENS.HARVARD.EDU
**Razieh Faghihpirayesh**[*1,2]                               RAZIEHFAGHIH@ECE.NEU.EDU
**Deniz Erdoğmuş**[2]                                         D.ERDOGMUS@NORTHEASTERN.EDU
**Ali Gholpour**[1]                                          ALI.GHOLIPOUR@CHILDRENS.HARVARD.EDU

[1] *Boston Children's Hospital and Harvard Medical School, Boston, MA*
[2] *Northeastern University, Boston, MA*

**Editors:** Accepted for publication at MIDL 2024

## Abstract

In this paper, we introduce a novel end-to-end predictive model for efficient fetal motion correction using deep neural networks. Diverging from conventional methods that estimate fetal brain motions and geometric distortions separately, our approach introduces a newly developed joint learning framework that not only reliably estimates various degrees of rigid movements, but also effectively corrects local geometric distortions of fetal brain images. Specifically, we first develop a method to learn rigid motion through a closed-form update integrated into network training. Subsequently, we incorporate a diffeomorphic deformation estimation model to guide the motion correction network, particularly in regions where local distortions and deformations occur. To the best of our knowledge, our study is the first to simultaneously track fetal motion and address geometric deformations in fetal echo-planar images. We validated our model using real fetal functional magnetic resonance imaging data with simulated and real motions. Our method demonstrates significant practical value to measure, track, and correct fetal motion in fetal MRI.

**Keywords:** Deep Learning, Fetal MRI, Rigid Motion Correction, Geometric Deformation.

## 1. Introduction

Motion estimation is a critical procedure, meticulously designed to correct image artifacts induced by object motion, especially in the realm of medical image analysis for fetal imaging. Its critical role spans across various fields, including image segmentation (Ebner et al., 2020; Faghihpirayesh et al., 2022), reconstruction (Gholipour et al., 2010; Kuklisova-Murgasova et al., 2012; Xu et al., 2023), and pose estimation (Salehi et al., 2018; Golland and Adalsteinsson, 2020), among others. Effective motion compensation and correction techniques significantly contribute to the overall accuracy and efficiency of fetal imaging studies (Malamateniou et al., 2013). Estimating fetal motion presents inherent challenges, notably the unpredictable nature of fetal head movements and local geometric distortions due to accumulated imaging errors. To overcome these challenges, existing motion correction methods mostly fall into two distinct categories: i) iterative optimization methods based on mathematical models, and ii) predictive models learned by deep neural networks.

---

[*] Contributed equally

One of the initial paradigms of 3D fetal brain reconstruction using motion correction was a three-step model (Rousseau et al., 2006) involving multi-resolution slice alignment for motion correction, intensity non-uniformity correction, and super-resolution reconstruction through scattered data interpolation. To address motion errors from 2D slice misalignments, a slice-to-volume registration (SVR) model was introduced (Jiang et al., 2007), incorporating a scattered data interpolation method with a novel multi-level B-spline kernel. A motion correction technique was presented in (Kim et al., 2009) to align image stacks based on 2D slice intersections to facilitate the reconstruction of a high isotropic resolution 3D volume. A forward model of slice acquisition and its inverse problem solution were proposed in (Gholipour et al., 2010). This approach featured a robust M-estimation solution to reduce the effects of corrupted (outlier) slices in a super-resolution reconstruction framework. Building upon these developments, an SVR approach was presented to encompass complete outlier removal through robust statistics based on the expectation-maximization algorithm (Kuklisova-Murgasova et al., 2012). Tourbier et al. (Tourbier et al., 2015) used total variation regularized SVR, solved using the primal-dual hybrid gradient method. While these advancements have enhanced the accuracy and efficiency of motion correction in image reconstruction, all gradient-based optimization methods for slice realignments have a very limited capture range, often failing in cases of large and rapid fetal movements.

Motivated by the progress in deep learning, a spectrum of models has emerged, with a particular focus on advancing predictive motion estimation (Mahendran et al., 2017). Hou et al. developed a method for predicting 3D rigid transformations of arbitrarily oriented 2D slices (Hou et al., 2018). This pursuit expanded with the introduction of a real-time fetal motion tracking system, leveraging the strengths of diverse neural networks, including Convolutional Neural Networks (CNNs) and Recurrent Neural Networks (RNNs), to predict motion directly from input images (Salehi et al., 2018; Singh et al., 2020; Evan et al., 2022).

To address non-rigid deformations in fetal MRI, two distinct approaches were presented. These include the patch-to-volume reconstruction technique (Alansary et al., 2017) and the deformable SVR technique (Uus et al., 2020), both designed to correct non-rigid motion to reconstruct deformable anatomy such as fetal body parts. In the domain of automated fetal MRI reconstruction, Ebner et al. developed a toolkit that featured slice-level outlier rejection, fetal brain localization, and volume reconstruction (Ebner et al., 2020).

Noteworthy progress has been achieved in the domain of sensorless imaging, particularly in 3D volume reconstruction from 2D freehand ultrasound images, leveraging deep implicit representations (Yeung et al., 2021). Xu et al. incorporated transformers (Vaswani et al., 2017) trained on synthetically transformed data, streamlining automatic relevance detection between slices (Xu et al., 2022). Advancing this approach, Xu et al. further introduced a reconstruction technique that incorporates implicit neural representations, enhancing image reconstruction performance (Xu et al., 2023). Evan et al. introduced KeyMorph, an unsupervised deep learning framework designed for robust and interpretable multi-modal medical image registration (Evan et al., 2022). Utilizing anatomically-consistent keypoints, KeyMorph aims to enhance the registration process and increase alignment accuracy. Moyer et al. introduced a real-time method for tracking rigid motion by employing equivariant neural networks (Moyer et al., 2021). The effectiveness of this approach in capturing significant rigid motions is derived from the intrinsic rotation-equivariant nature of equivariant filters (Cohen and Welling, 2016).

Despite the notable performance achieved by the aforementioned methods, there is currently a lack of an end-to-end predictive motion correction framework that can adequately address both uncontrollable fetal motion and geometric distortions. This gap in the existing literature motivated our endeavor to develop a comprehensive framework proficient in efficiently managing fetal motions and geometric distortions through deep learning. The core contributions of our proposed motion correction method are summarized in three folds:

- **Pioneering contribution**: Our study introduces the first comprehensive end-to-end predictive motion tracking framework adept at handling both significant rigid motion and geometric distortions in fetal echo planar imaging (EPI).

- **Significance**: Our proposed framework not only achieves motion correction accuracy comparable to current state-of-the-art methods but also ensures faster and more stable convergence. This attribute is of significant value, especially in real-time, automated fetal head steering systems, where frequent and substantial motions occur.

- **Theoretical advancement for broad applicability**: The theoretical tool developed through our joint learning algorithm demonstrates wide applicability, offering advantages in various fetal imaging applications. This includes real-time fetal brain segmentation, facilitating population-based studies of fetal brain development using mean template estimation, and improving EPI image reconstruction with advanced distortion correction techniques.

## 2. Methodology

In this section, we first outline the theoretical basis needed to address rigid motions and deformations in fetal imaging. We start by reviewing the principles of rigid motion estimation. Next, we focus on correcting geometric distortions, specifically using Large Deformation Diffeomorphic Metric Mapping (LDDMM), chosen for its effectiveness in creating smooth, invertible mappings that maximally maintain the correctness of topological information (Beg et al., 2005). Finally, we describe the design of our proposed network architecture and the approach to its training.

### 2.1. Rigid Motion Estimation

Rigid motion estimation aims to identify the best translation $\mathcal{T}$ and rotation $\mathcal{R}$ parameters that define a rigid transformation $Q(\mathcal{T}, \mathcal{R})$ between a pairwise images. This process involves minimizing the Euclidean distance $d$ between the source image $S$ and the target image $T$,

$$Q^*(\mathcal{T}, \mathcal{R}) = \arg\min_{Q} \ d[S \circ Q(\mathcal{T}, \mathcal{R}), T], \tag{1}$$

where $\circ$ is the composition operator that resamples $S$ using the rigid transformation $Q(\mathcal{T}, \mathcal{R})$. When this operator is applied to any vector $\mathbf{v}$, it yields a transformed vector $Q(\mathbf{v})$ of the form $Q(\mathbf{v}) = \mathcal{R}\mathbf{v} + \mathcal{T}$. Here, $\mathcal{R}^T = \mathcal{R}^{-1}$ indicating that $\mathcal{R}$ is an orthogonal matrix.

The transformation function $Q$ can be calculated using low-dimensional representations, such as point clouds or key points of images (Moyer et al., 2021; Evan et al., 2022). This

process is quantified by the energy function $E(Q)$, as shown in the equation,

$$E(Q) = \|S \circ Q(\mathcal{T}, \mathcal{R}) - T\|_F, \tag{2}$$

where $\|\cdot\|_F$ denotes the Frobenius norm, and $\bar{S}, \bar{T}$ represent the low-dimensional representations of the source and target respectively. We derive the closed-form solution for both translation and rotation parameters,

$$\mathcal{T} = \bar{T} - \mathcal{R}\bar{S}, \quad \mathcal{R} = V \cdot U^T, \quad \text{s.t. } \det(\mathcal{R}) = 1, \tag{3}$$

where $U\Sigma V^* = \bar{S} \cdot \bar{T}^T$, $U$ and $V^*$ are real orthogonal matrices, $\Sigma$ is a diagonal matrix with non-negative real numbers on the diagonal. Setting the determinant of $\mathcal{R}$ to 1 guarantees that it accurately reflects a rigid transformation.

## 2.2. Geometric Deformation Correction via LDDMM

In this section, we provide an overview of the LDDMM algorithm for image registration (Beg et al., 2005). This algorithm is employed to address deformable geometric distortions between the rigid motion-corrected image $S \circ Q(\mathcal{T}, \mathcal{R})$ and the target image $T$. For simplicity, we denote $\hat{\mathbf{S}}$ as $S \circ Q(\mathcal{T}, \mathcal{R})$.

Given both $\hat{\mathbf{S}}$ and $T$, defined on a $d$-dimensional torus domain $\Gamma = \mathbb{R}^d/\mathbb{Z}^d$ ($\hat{\mathbf{S}}(x), T(x) : \Gamma \to \mathbb{R}$), the objective of diffeomorphic image registration is to find the shortest path to generate time-varying diffeomorphisms $\{\psi_t(x)\} : t \in [0, 1]$ such that $\hat{\mathbf{S}} \circ \psi_1$ is similar to $T$. This is typically solved by minimizing an explicit energy function over the initial transformation fields $v_0$ (Vialard et al., 2012), which can be expressed as follows,

$$E(v_0) = \frac{1}{\sigma^2}\|\hat{\mathbf{S}} \circ \psi_1(v_0) - T\|_2^2 + (Lv_0, v_0), \tag{4}$$

where $\sigma^2$ is noise variance in images and $L : V \to V^*$ is a symmetric, positive-definite differential operator that maps a tangent vector $v_t \in V$ into its dual space as a momentum vector $m_t \in V^*$. This is typically denoted as $m_t = Lv_t$ or $v_t = Km_t$, with $K$ being an inverse operator of $L$. The notation $(\cdot, \cdot)$ denotes the pairing of a momentum vector with a tangent vector, which is similar to an inner product.

The geodesic shooting process states that the geodesic path $\{\psi_t\}$ can be uniquely determined by integrating a given initial velocity $v_0$ forward in time by using the Euler-Poincaré differential (EPDiff) equation (Arnol'd, 1966; Miller et al., 2006) as:

$$\frac{\partial v_t}{\partial t} = -K\left[(Dv_t)^T \cdot m_t + Dm_t \cdot v_t + m_t \cdot \operatorname{div} v_t\right], \quad \frac{d\psi_t}{dt} = -D\psi_t \cdot v_t, \tag{5}$$

where the operator $D$ denotes a Jacobian matrix and $\cdot$ represents element-wise matrix multiplication. Here, div is the divergence.

## 2.3. Network Architecture and Training

We develop a deep learning model that explicitly estimates both rigid motion and geometric distortions by integrating a combination of Eq. (2) and Eq. (4) into our objective function. We show that such joint estimation results in improved accuracy and robustness of the

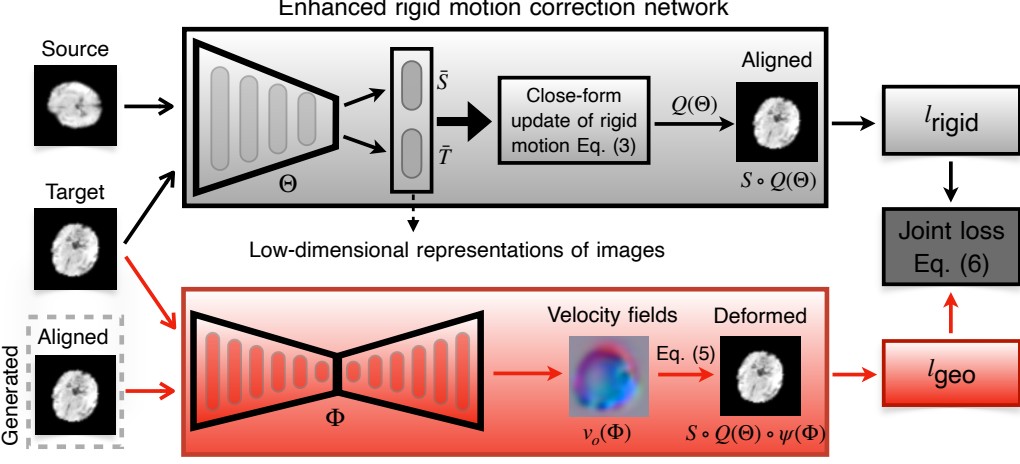

Figure 1: An illustration of our proposed **joint correction network (JCN)** architecture. The top module performs rigid motion correction on pairs of images using closed-form updates derived from their key point representations. The bottom module corrects geometric distortions in the aligned and target images, employing a geometric loss function with a regularization term in diffeomorphic transformation space. Both models are interrelated—the geometrically corrected data is fed back as augmented data to enhance the accuracy of rigid motion correction. This, in turn, aligns better with the geometric distortion network, facilitating the correction of local distortions in corresponding positions of the fetal brain.

model. Our framework consists of two modules: an enhanced rigid motion correction neural network that corrects rigid motions with closed-form update, and an unsupervised learning of geometric distortion correction through an image registration network. An illustration of our joint learning framework is shown in Fig. 1. Below, we provide a detailed description of our network architecture and objective function.

**Enhanced motion correction network.** Let $\Theta = (\mathcal{T}, \mathcal{R})$ represent the encoder parameters that learn rigid parameters and key features from image spaces, with $Q(\Theta)$ denoting the transformation function yielded by the learned rigid parameters from the key points of images. The rigid correction loss is computed between the aligned and target images. $\Theta$ is characterized using one of two backbone structures, (i) a 7-layer equivariant neural network to effectively capture rotation-equivaraint representations using 3D steerable CNNs (Weiler et al., 2018; Moyer et al., 2021), and (ii) a conventional 9-layer CNN as employed in Key-Morph (Evan et al., 2022).

**Geometric distortion correction based on image registration network.** Let $\Phi$ denote the parameters of an encoder-decoder in our geometric learning network, where $\psi(\Phi)$ represents the deformation fields and $v_0(\Phi)$ represents the velocity fields learned by the network. Except for estimating deformations by LDDMM (Hinkle, 2018), we also provide

another deep learning model that learns stationary velocity fields (Balakrishnan et al., 2019) meanwhile maintains a comparable model accuracy. Advanced predictive image registration models, including TransMorph (Chen et al., 2022) and DiffuseMorph (Kim et al., 2022), can easily be integrated into our framework.

**Joint correction network and objective function.** Our joint correction network (JCN) comes in two versions: JCN: EF, which employs equivariant filters, and JCN: KM, based on the CNN architecture that is similar to KeyMorph. Our total objective function, which serves as the network's training loss, combines both the motion correction component (Eq. (2)) and the geometric deformation estimation (Eq. (4)), formulated as follows,

$$l(\Theta, \Phi) = \underbrace{\lambda \|S \circ Q(\Theta) - T\|_F}_{l_{\text{rigid}}} + \underbrace{\frac{1}{\sigma^2} \|S \circ Q(\Theta) \circ \psi(\Phi) - T\|_2^2 + (Lv_0(\Phi), v_0(\Phi))}_{l_{\text{geo}}}$$
$$+ \text{reg}(\Theta, \Phi), \qquad \text{s.t. Eq. (3), \& Eq. (5).} \tag{6}$$

Here, $\text{reg}(\cdot)$ is a regularization term constrained on network parameters, and $\lambda$ is a weighting factor balancing the effects of both networks. Other image dissimilarity terms, such as normalized cross correlation (Avants et al., 2008) and mutual information (Wells III et al., 1996; Wang et al., 2023) can be adopted in our framework.

## 3. Experimental Setup

**Data.** Our study includes $1,881$ pairs of 3D EPIs from fMRI time series of 15 subjects who underwent fetal MRI scans using a Siemens 3T scanner (Skyra or Prisma) between August 2015 and September 2021. Approved by the institutional review board, the study obtained written informed consent from all participants. The dataset covers a gestational age range from 22.57 to 38.14 weeks (mean 32.39 weeks). Imaging parameters included a slice thickness of 2 to $3mm$, TR of 2 to 5.6 seconds (mean 3.1 seconds), TE of 0.03 to 0.08 seconds (mean 0.04 seconds), and a FA of 90 degrees. Fetal brains were extracted using a real-time deep learning segmentation method (Faghihpirayesh et al., 2023), resampled to $96^3$ with a voxel resolution of $3mm^3$, and underwent intensity normalization.

**Baselines & Evaluation Metrics.** We compared various motion correction methods, including conventional iterative rigid registration in ITK to estimate rigid transformation as 3D versors (Ibanez et al., 2003), and rigid motion estimation by deep learning methods such as DeepPose (Salehi et al., 2018) and KeyMorph (Evan et al., 2022). We compared their performance with our joint models JCN: KM and JCN. To demonstrate the benefit of our joint learning, we report the best Dice score of disjoint approaches, where rigid motion estimation is treated as a preprocessing step before geometric distortion estimation. Performance was evaluated through visual comparison and by computing translational and angular errors on simulated motions. We also manually added translations (in mm) and rotations (in degrees) to real fMRI scans with natural fetal motions, altering their maximum motion values in three directions. To better demonstrate our model's stability, we compared the Dice coefficient by computing the overlapping regions between the aligned image and the target across three levels of motion[1].

---

1. Our code is released online, https://github.com/bchimagine/JointMotionTracking

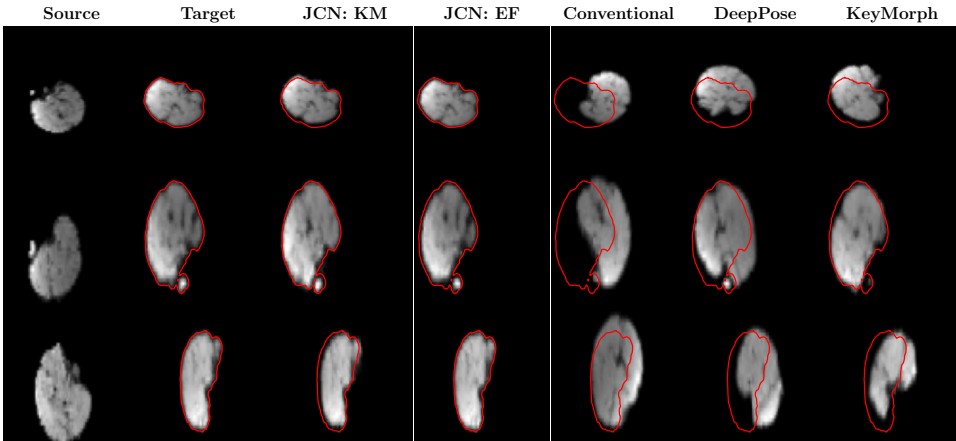

Figure 2: Fetal motion correction visualizations for all methods. Left to right: source, target (the contour is highlighted as red), motion-corrected images by our methods (JCN: KM and JCN: EF), the iterative conventional method, DeepPose, and KeyMorph. **Dice scores**, **0.95 for JCN: KM**, **0.97 for JCN: EF**, 0.32 for conventional, 0.63 for DeepPose, and 0.73 for KeyMorph. Brief Dice score comparison between the best model of disjoint ones and ours, 0.90/ **0.97**. Motion correction efficiency with average time consumption: **0.491s** per pair.

**Hyperparameters.** We set the dimension for low-dimensional key points to 128 when computing the close-form update of rigid transformation in Eq. (3) and set $\alpha = 3$ for the operator $L$ with using 5 as time steps of Euler integration for geodesic shooting (Eq. (5)). The noise variance was fixed at $\sigma = 0.02$. For network training, we set batch size as 4, weight decay as 0.0001 for a $L_2$ regularization, and a learning rate of $\eta = 1e-5$ for 300 epochs with Adam optimizer. We divided the data into training (1283 pairs from 11 subjects), validation (299 pairs from 2 subjects), and testing (299 pairs from 2 subjects). The best-performing networks were saved based on validation performance across all models. All experiments were conducted with an NVIDIA RTX A6000 GPU.

## 4. Results

Fig. 2 presents a motion correction study for pairwise fMRI scans characterized by significant motions. It shows that the conventional and DeepPose methods fail to correctly align images, while our approach excels, closely matching the target image. Our method effectively handles both rigid motion and local geometric deformations, leveraging estimated velocity fields for more accurate correction.

Fig. 3 illustrates the comparative analysis of motion correction errors in translation and rotation across different methods. It highlights the enhanced performance of our proposed technique in correcting fetal motions, surpassing the state-of-the-art methods.

Fig. 4 shows the dice coefficient comparisons for fMRI time sequences under various motion conditions. It demonstrates that our method consistently achieves higher dice scores,

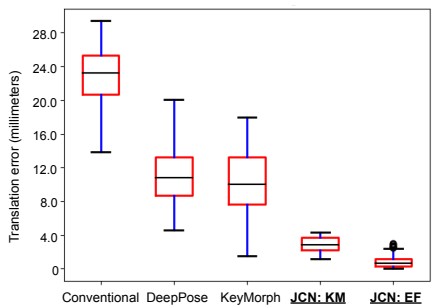 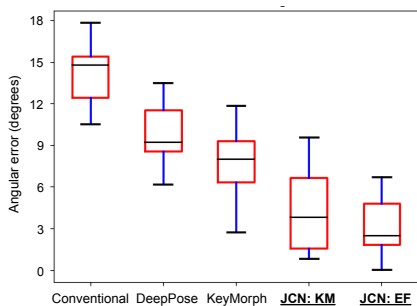

Figure 3: Motion correction comparison of translation and rotation errors (averaged for all directions) of 299 real fMRI scans with simulated motions.

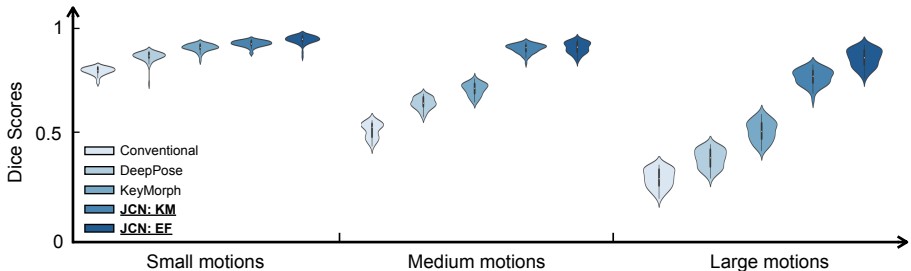

Figure 4: Motion correction performance by dice evaluation on real fMRI with different degrees of motions, small ($\mathcal{T}_{max} = 10$mm, $\mathcal{R}_{max} = 5°$), medium ($\mathcal{T}_{max} = 20$mm, $\mathcal{R}_{max} = 10°$) and large motions ($\mathcal{T}_{max} = 30$mm, $\mathcal{R}_{max} = 20°$). The dice score of our best model, for motion levels from left to right are, 0.97, 0.93, 0.92.

regardless of the motion degree. This highlights the robustness and stability of our model, particularly in challenging scenarios with significant motion occurrences.

## 5. Conclusion

This paper presents a pioneering predictive model for fetal motion correction using deep neural networks. In contrast to conventional approaches that independently estimate fetal brain motions and geometric distortions, our proposed method adopts an efficient and robust joint learning framework. This framework excels in achieving optimal and consistent performance across various degrees of fetal motions. The model demonstrates notable effectiveness on fMRI data with both simulated and real motions, showcasing significant potential in real-time tracking and steering systems for fetal head motion. An intriguing avenue for future exploration involves applying our developed model to image reconstruction for EPI data, which poses more challenging fetal motion artifacts.

## Acknowledgments

This research was supported in part by the National Institute of Biomedical Imaging and Bioengineering, the National Institute of Neurological Disorders and Stroke, and Eunice Kennedy Shriver National Institute of Child Health and Human Development of the National Institutes of Health (NIH) under award numbers R01NS106030, R01EB031849, R01EB032366, and R01HD109395; and in part by the Office of the Director of the NIH under award number S10OD025111. This research was also partly supported by NVIDIA Corporation and utilized NVIDIA RTX A6000 and RTX A5000 GPUs. The content of this publication is solely the responsibility of the authors and does not necessarily represent the official views of the NIH or NVIDIA.

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
