# OpenReview forum: "Joint Motion Estimation with Geometric Deformation Correction for Fetal Echo Planar Images Via Deep Learning"
_MIDL.io/2024/Conference — MIDL 2024 Poster_

### Official Review · Reviewer_7jkf · 2024-02-28

**Confidence:** 4
**Preliminary Rating:** 5
**Recommendation:** Oral
**Final Rating:** 5

**Summary:**

This paper presents an algorithm that mixes rigid and non-rigid transformations to deform fetal MRI images for motion correction. The use of rigid transformation and LDDMM based deformations together in a deep learning setup is interesting, and seems to yield impressing results, as compared to previous methods.

**Strengths:**

The paper is well written, with a clear exposition of the problem, methodology, datasets and results. The exposition of the LDDMM framework is concise but sufficient for non-expert readers. The results and the comparison with the previous method are clear and show well the strength of this approach.

**Weaknesses:**

I did not see any major weaknesses in this work. What I could suggest as improvements would be a more extensive survey of recent papers on LDDMM and ML. Only a few are cited on this topic, and i think there are more relevant works out there that could be interesting to relate to this one.

**Detailed Comments:**

I would only suggest the authors to add a few more relevant refs on LDDMM with machine learning, if they find any of interest, to put this work a little more into this context.

**Justification Of Final Rating:**

Very good, I do not have any further comments.                                                                                                                                                            .

**Justification Of The Preliminary Rating:**

This is a nice piece of work, combining a now classical LDDMM framework with deep learning, that produces very decent results, applicable on a wide range of topics. I think that this work deserves to be presented to foster discussions on this topic.

**Questions To Address In The Rebuttal:**

I don't have any pressing questions to be answered.

**Special Issue:**

Yes

---

> ### Author Response · Authors · 2024-03-14
>
> We greatly thank the positive feedback from the reviewer! We will conduct a thorough review of existing deformable image registration models based on deep neural networks, such as TransMorph [1] and Diffusemorph [2]. We will also enrich the literature review of  motion tracking in both the medical image analysis and the computer vision domain in the extended journal version of our paper.
>
> **References**
> [1]. Chen, Junyu, et al. "Transmorph: Transformer for unsupervised medical image registration." Medical image analysis 82 (2022): 102615.
> [2]. Kim, Boah, Inhwa Han, and Jong Chul Ye. "Diffusemorph: Unsupervised deformable image registration using diffusion model." European conference on computer vision. Cham: Springer Nature Switzerland, 2022.

---

### Official Review · Reviewer_TFsQ · 2024-02-28

**Confidence:** 4
**Preliminary Rating:** 4
**Final Rating:** 2

**Summary:**

The manuscript presents a method for fetal pose estimation. The method combines rigid and deformable registration in a jointly optimizing end-to-end architecture. It is compared to a number of existing methods, using a dataset of 15 subjects for training and evaluation.

**Strengths:**

The authors present a sensible and well-described method. The descriptions are generally easy to follow and complete.

The visualization in figure 1 is very strong; it provides an immediate and intuitive explanation for how the method works.

**Weaknesses:**

The authors state "We show that such joint estimation results in improved accuracy and robustness of the model", but I do not think they adequately show this in the results. The rigid model-only (i.e. without the subsequent deformable model and loss) is not added as a baseline, so it is hard to tell whether the joint loss caused the strong performance, or whether this is simply the result of a well-tuned network for rigid motion estimation.

The method is trained and evaluated on a large number of image pairs from 15 subjects. However, it is not clear how the train/test split was done. The authors state they used 15% of the data for evaluation, but 15% of 15 subjects is 2.25, which is impossible. The authors should clarify whether they split on a subject-level, or split on an image pair-level.

The comparison of dice overlap seems inappropriate to me, as the authors compare their deformable registration model with rigid and affine ones. It would make more sense to compare to a baseline pipeline where the rigid model is trained first, and the deformable network is trained after (i.e. not end-to-end and without a joint loss), and both are applied subsequently. It is extremely common to perform affine or rigid pre-alignment before applying a deformable registration method, so this would match the typical application of such methods.

**Detailed Comments:**

The description of the conventional method seems to be missing. Which one was used here?

**Justification Of Final Rating:**

The authors have unfortunately not added the separate rigid correction network as a baseline in their revision. I would argue this is the most important baseline of all, as the joint optimization of the two networks is the main novelty of the work. Hence, it is not possible to judge the benefit of their contribution from the current version of the manuscript. Additionally, the authors have not updated the visualization in figure 4 to do an apples-to-apples comparison, still comparing Dice scores of their deformable method with Dice scores of rigid transformations. This is not an appropriate representation of the state of the art and misrepresents the benefit of the proposed method.

The work clearly has potential, but the current version of the manuscript is not in a publishable state. A number of problems with the paper have been addressed in the OpenReview comments to both my own and the other reviews, but readers should not be required to search for the OpenReview page in order to get the full picture of the work. Hence, I unfortunately have to downgrade my initial assessment.

**Justification Of The Preliminary Rating:**

The presented method seems sensible and is well described. However, the method evaluation is not very strong, which makes it difficult to confirm whether the attained performance increase is truly a result of the joint loss and end-to-end approach.

**Questions To Address In The Rebuttal:**

See: the points in weaknesses

---

> ### Author Response · Authors · 2024-03-14
>
> We greatly thank the reviewer for the comments and feedback on our manuscript! Below, we present detailed responses to each point raised.
>
> **1. Dataset and Baselines**  We appreciate the reviewer's attention to detail regarding the dataset and baselines. In response, we will enhance the clarity of our manuscript by explicitly describing the dataset split—comprising training (1283 pairs from 11 subjects), validation (299 pairs from 2 subjects), and testing (299 pairs from 2 subjects). We used an iterative rigid image registration model, as conventional one, for estimating 3D versors, encompassing both translation and rotation. This model, implemented in C++ using ITK, will be clarified in the revised manuscript.
>
> **2. Joint Learning** To address the reviewer's concern, we would like to clarify that we show the results of models that focus solely on rigid motion estimation. For instance, we have KeyMorph, DeepPose for rigid motion only, and JCN-based models that include joint geometric distortion correction (Fig.2 ~ 4). Both models are interrelated—the geometrically corrected data is fed back as augmented data to enhance the accuracy of rigid motion correction. This, in turn, aligns better with the geometric distortion network, facilitating the correction of local distortions in corresponding positions of the fetal brain. We will clearly depict this mutually beneficial relationship in our main figure (Fig. 1). We report the accuracy of both joint and disjoint approaches here. The dice evaluation results for disjoint approaches (best performance on rigid and geometric distortion) are 0.843/0.905, while for our joint approaches (best performance on rigid and geometric distortion), the results are 0.912/0.971.
>
> We are willing to incorporate  clarifications and results above  into the revised manuscript.

---

> > ### Comment · Reviewer_TFsQ · 2024-03-19
> > **Wrong pdf uploaded?**
> >
> > Looking at the revised manuscript, I don't see any mention or description of the disjoint baseline, nor any description of the original "conventional" baseline. Could it be that you have uploaded the wrong pdf file?

---

> ### Author Response · Authors · 2024-03-19
> **Clarifications.**
>
> **Conventional method.**
> The description of the iterative conventional model was included in the Baselines of Experiment section.
>
> **Disjoint approaches.**
> Sorry for the confusion. As addressed in the previous response, we intended to report the quantitative results here. Now, we have added the brief comparison in the new manuscript as well. A more comprehensive analysis and visualization between the two types of models will be conducted in an extended paper of this work.

---

### Official Review · Reviewer_WKwa · 2024-02-29

**Confidence:** 4
**Preliminary Rating:** 5
**Recommendation:** Oral
**Final Rating:** 5

**Summary:**

This paper proposes a deep learning framework for jointly tracking rigid motion corruption and geometric distortion between slices in fetal EPI. The framework is comprised of two modules: one that estimates rigid motion parameters from key points detected from source and target images and another that estimates deformation and velocity fields characterizing the geometric distortion from the rigid motion-adjusted source and target images. The networks are trained jointly via a loss that combines (1) a loss encouraging the rigidly transformed source to match the target and (2) an energy function over the distortion transformation fields. The proposed method is tested on fetal EPI time series data with real and simulated fetal movements and is compared to rigid image registration as well as several deep learning baselines.

**Strengths:**

- The paper tackles a very challenging problem
- The paper is extremely clearly written and easy to follow
- The experimental results are compelling, compared to fair baselines, and include results on real data

**Weaknesses:**

- There is a small missing ablation study and metrics that are not computed in the current version of the draft that would further improve the paper
- Paper is missing commentary on the computational cost of the geodesic shooting
- Overall, these are small weaknesses and I do not believe these are critical for acceptance.

**Detailed Comments:**

- What is the regularizer used in Equation (6)?
- In Figure 1, I believe the Equation (5) label is not needed over the first arrow, which shows the output of the network, and is only needed over the second arrow, where the deformation fields are computed.
- Please comment on the computational cost of performing the geodesic shooting as part of the inference procedure; how much time does this require? Assuming this depends on the number of steps, how sensitive is the method to the number of steps used?
- To understand what the algorithm is doing, it would be helpful to visually show and quantitatively characterize an intermediate result: the rigidly aligned output of the motion correction network, before this is passed to the geometric deformation correction network. When this is computed for JCN:KM and compared to the KeyMorph result, it will separate the improvement from applying the the joint training strategy to the KM backbone from the improvement gained from the geometric deformation correction network. Similarly, when compared to the result of the geometric deformation correction network, it will show how the deformation correction network is helping after the rigid correction is complete.
- Does Figure 3 combine the translation errors in all axes and the angular errors in all axes?
- While the Dice coefficient captures the similarity between the shapes of the warped source and target, it does not capture similarities in the intensity structure/content within the shape. Reporting a supplemental metric like structure-level Dice or surface distance or similar may help solidify confidence that this information is accurate after tracking as well.

**Justification Of Final Rating:**

I was originally at strong accept and keep my rating, as the authors implemented all my questions/changes with the exception of visualizing the intermediate output of the rigid-correction network that was trained jointly with the deformation network.

This is related but not identical to reviewer TFsQ’s suggestion to visualize a rigid correction-only network. As mentioned in the authors’ rebuttal, the KeyMorph and DeepPose models are already presented as baselines, which I believe is sufficient for showing the contribution relative to rigid-only models. My suggestion to show the intermediate output of the jointly trained model would provide an additional layer of understanding when compared to the already present rigid-only baselines in the paper by showing how the joint training affects the rigid components. I agree with the authors that this is fine to leave for a journal extension.

Reviewer TFsQ’s suggestion to show the Dice losses of the separately and jointly trained networks is a good one, and I believe is now incorporated in Figure 2 caption, though not in the Dice plots summarized across the dataset. I think this is fine as an ablation study for now but would encourage the authors to make this limitation clearer and mention the ablation results in the main body, not just the figure captions.

**Justification Of The Preliminary Rating:**

I did not identify any major weaknesses, as the results are compelling, the method is clear and intuitive, and the comparisons are to solid baselines. Further, this is a challenging problem. I thus recommend acceptance/oral presentation.

**Questions To Address In The Rebuttal:**

- What is the regularizer used in Equation (6)?
- Does Figure 3 combine the translation errors in all axes and the angular errors in all axes?

**Special Issue:**

Yes

---

> ### Author Response · Authors · 2024-03-14
>
> We thank the valuable feedback from the reviewer and carefully address each point as follows,
>
> **1. Regularization.**
> We incorporate weight decay as L-2 regularization during the training of our network.
>
> **2. Computational Efficiency.**
> Thanks for highlighting the importance of computational efficiency. The computational time for motion estimation on average of testing data is 0.491 seconds per 3D pair (96^3). We agree that larger integration steps can impact computational time. In our joint learning framework, we've optimized our approach by using a fast version of LDDMM (specifically, lagoMorph[1] on GPU) with the time steps parameter set to 5. Additionally, we will also provide another fast shooting method with stationary velocity fields that maintains a same-level of accuracy in our code.
>
> **3. Result Visualization.**
> It is a great suggestion to display intermediate results between two joint models. We will carefully address this and add clear visualization of geometric distortion correction process by presenting velocity fields in the experiment section of an extended journal version of this work. The title of Fig. 3 will be revised to ensure clarity regarding the inclusion of average translation and angular errors for all directions.
>
> **4. Improvements of Evaluation.**
> We agree with the reviewer's suggestion regarding structural-wise or voxel-wise evaluation metrics. We plan to develop such metrics and conduct a thorough evaluation in an extended version of the journal paper dedicated to this work.
>
> **References**
> [1]. Hinkle, Jacob D. Lagomorph. No. Lagomorph. Oak Ridge National Laboratory (ORNL), Oak Ridge, TN (United States), 2018.

---

### Comment · Area_Chair_dyu4 · 2024-03-18
**Please implement the promised changes**

I would like to remind the authors that, this year, the revised manuscript should be submitted along with the responses to the reviewers. It is better to highlight these changes in red for improved readability. Unless I missed something, it seems like this is not the case for this submission. The camera-ready manuscript will be submitted later, upon acceptance.

---

> ### Author Response · Authors · 2024-03-18
>
> Thanks for your kind reminder! We will upload the revised manuscript ASAP.

---

### Meta-Review · Area_Chair_dyu4 · 2024-03-29

**Recommendation:** Accept (Poster)
**Confidence:** 5

**Metareview:**

Since the reviewers had mixed feelings about this paper, I carefully read it as well as the rebuttal.

First, a number of issues have been answered by the authors, but only on OpenReview not in the paper, which is not ideal for readers.

Then, I agree with reviewer TFsQ (and to a lesser extent WKwa) that the evaluation of this method lacks clear ablations. Comparing against different rigid methods is interesting but not central, since the authors did not propose a new rigid framework, but rather took existing ones and incorporated them into their joint optimisation. Here, since the main contribution is the joint optimisation of rigid/elastic registration, the main experiment should compare each step against models separately trained for rigid/elastic registration. The authors have done a small experiment, but the results are only included in the caption of a visual result figure, which is not enough in my opinion. I also think the figure suggested by reviewer WKwa should have been included, at least in the Appendix.

Also, there are some important missing/unclear information: How do the authors derive equation (3) ? What is the architecture of the non-linear deformation network ? What are the rationales behind the 2nd term in (4) ?

Finally, I also spotted some notation problems: $\mathcal{R}$ and $\mathcal{T}$ are directly computed from $\bar{S}, \bar{T}$ so I don’t think $Q$ should depend on $\bar{S}, \bar{T}$ anymore. $V$ is used for different variables in sections 2.1/2.2. Then, in section 2.3 $\Theta$ both represents the encoder parameters and its outputs $\Theta=(\bar{S}, \bar{T}, R, T)$. $\alpha$ has not been properly introduced.

Overall, I recommend to accept this paper, because it tackles an important problem with wide applications (not restricted to fatal imaging) and is likely to foster interesting discussions at MIDL. However, I think the evaluation flaws are too fatal to recommend an oral presentation. The authors are encouraged to include as many changes as possible in the final version.

---

### Decision · Program_Chairs · 2024-04-06

Accept (Poster)